# Using Unity to Help Solve Reinforcement Learning

**Connor Brennan**\*
Mila–Quebec AI Institute
Université de Montréal

**Andrew Robert Williams**\*
Mila–Quebec AI Institute
Université de Montréal

**Omar G. Younis**†
Mila–Quebec AI Institute
Université de Montréal

**Vedant Vyas**†
Department of Computing Science
University of Alberta

**Daria Yasafova**
Mila–Quebec AI Institute
Université de Montréal

**Irina Rish**
Mila–Quebec AI Institute
Université de Montréal

## Abstract

Leveraging the depth and flexibility of XLand as well as the rapid prototyping features of the Unity engine, we present the United Unity Universe, an open-source toolkit designed to accelerate the creation of innovative reinforcement learning environments. This toolkit includes a robust implementation of OpenXLand, a framework for meta-RL based on XLand 2.0 [23], complemented by a user-friendly interface which allows users to modify the details of procedurally generated terrains and task rules with ease. Along with a ready-to-use implementation of OpenXLand, we provide a curated selection of terrains and rule sets, accompanied by implementations of reinforcement learning baselines to facilitate quick experimentation with novel architectural designs for adaptive agents. Furthermore, we illustrate how the United Unity Universe serves as a high-level language that enables researchers to develop diverse and endlessly variable 3D environments within a unified framework. This functionality establishes the United Unity Universe (U3) as an essential tool for advancing the field of reinforcement learning, especially in the development of adaptive and generalizable learning systems.

## 1 Introduction

Reinforcement learning (RL) and meta-reinforcement learning (meta-RL) foundation models have shown potential as adaptive agents that quickly adapt to diverse tasks in open-ended settings [23]. However, open-source options for open-ended environment generation rarely go beyond constrained subsets of scenarios, with limited control or open-endedness. The availability of frameworks for diverse, controllable, and extendable environment generation would be a boon to the development of generally adaptive agents and meta-RL in general.

Inspired by the use of the Unity Game Engine for reinforcement learning development [27, 24, 23], we propose the *United Unity Universe* (U3), an open-source environment development ecosystem based on the Unity game engine, which is freely available for personal use.[1]

---

\*Equal contribution.
†Equal contribution.
[1]https://unity.com/products/unity-personal

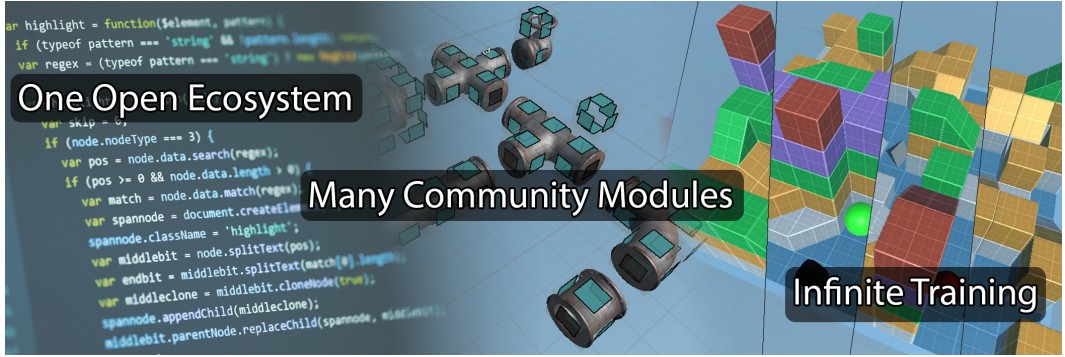

Figure 1: **United Unity Universe.** U3 is a single open-sourced ecosystem with which the machine learning community can easily construct novel 3D environments without sacrificing fine-grain control over the task or the world. U3 provides a high level language to easily define rules and procedurally generate new worlds layouts, resulting in effectively infinite variety, even within a single environment.

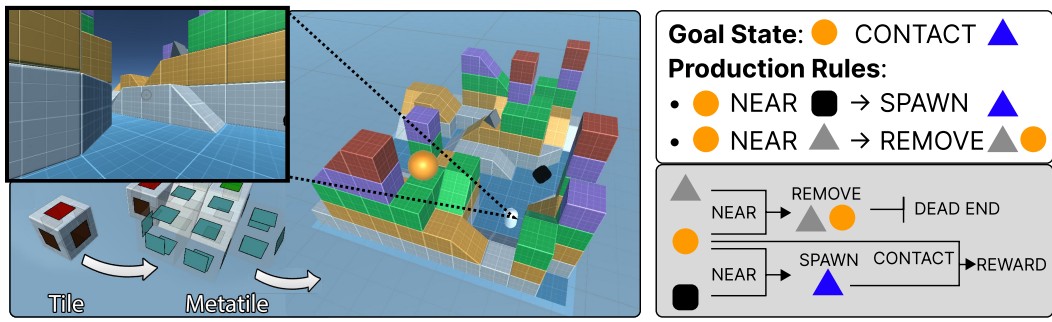

Figure 2: **An example of world generation with U3.** We use U3 to create OpenXLand, our open-source implementation of the XLand environment space for meta-reinforcement learning [24, 23]. **Bottom left**: tiles make up metatiles, which we can sample to procedurally generate 3D worlds, within which an agent acts. **Right**: U3 can also procedurally generate tasks: we sample objects and conditions on the objects that define the rules of a task, with a final goal state that rewards the agent. In this environment, the agent receives reward when the orange sphere makes contact with the blue pyramid. We see that the orange sphere is elevated, and therefore the agent must find it and use the ramps to access it. As to the blue pyramid, we do not see it because it is not there: the agent must first get the orange sphere near the black rounded cube to spawn the blue pyramid. This environment also contains a grey pyramid that serves as a distraction. Importantly, if the agent brings the grey pyramid near the orange sphere, both will disappear, making it impossible for the agent to spawn a blue pyramid and subsequently obtain its reward. **Top left:** The agent's perspective at initialization. The agent does not see the orange sphere, and so will have to search for it. However, the grey pyramid, which could lead to a dead end, is visible. As pointed out in [23], tasks in OpenXLand require many skills such as navigation, exploration, experimentation, and avoiding irreversible transitions. However, since the innumerable tasks are composed of a small number of components, the agent can improve performance by learning to reuse previously acquired knowledge in new tasks.

U3 uses procedural generation in a controllable and extendable fashion, making available an endlessly diverse set of environments for RL and meta-RL. U3 can also be scaled to many instances for efficient training. We use the U3 ecosystem to implement *OpenXLand*, an open-source, open-ended RL framework based on the XLand line on work [24, 23]. We make available six million pre-generated and filtered terrains, along with six million pre-generated tasks as datasets of varying difficulty, for a total 36 trillion unique combinations. We also include a pipeline between U3 and Python based on the Unity ML-Agents Toolkit [14], with clean implementations of RL algorithms such as Soft Actor-Critic (SAC) [11] and sidechannel support for dynamic readjustment of the generated environments or their generation processes.

**Concretely, our main contributions are:**

- **United Unity Universe (U3), an open-source Unity-based environment development ecosystem.**
    - Extendable world and task generation processes for rich environmental dynamics, as well as direct access to free Unity assets for building real-world environments.
    - A two-way pipeline connecting U3 and Python for efficient training, agent evaluation and runtime adaptive auto-curricula.
    - Scalable environments that can be run on multiple CPUs simultaneously.
- **OpenXLand, an XLand-style environment space of 3D environments in Unity with rich dynamics and compositional tasks.**
    - A 3D world procedural generation system that makes an XLand-style environment space available to the reinforcement learning community.
    - A production rule procedural generation system that extends XLand 2.0 with conjunctions of predicates, additional actions and additional objects.
    - Six million OpenXLand datasets and production rules for easy comparison across models over 36 trillion possible environment combinations.

Section 2 contains related work for environment procedural generation, including the main line of Unity-based meta-RL work inspiring U3 and OpenXLand. In Section 3, we cover the requisite background, particularly on Unity-based environments for RL [27] and XLand [24, 23]. We then go over U3 in Section 4, focusing on its use of procedural generation, scalability and extendability. Section 5 describes our OpenXLand implementation, its datasets and implemented RL baselines. Section 6 reviews current limitations, and we conclude in Section 7.

## 2 Related Work

### 2.1 Meta-Reinforcement Learning

The fast adaptation of foundation models has garnered attention recently as "in-context learning" [7]. Significant efforts have been directed toward creating environments to study fast adaptation in the context of meta-RL. Training agents to continuously learn across an open-ended distribution of task spaces requires such an open-ended collection of environments. Frameworks that generate collections provide a tradeoff between collection size and diversity. Dennis et al. [6] put forward the formulation of Unsupervised Environment Design, where underspecified RL environments can generate diverse RL environments by instantiating unspecified parameters. For example, environment generation frameworks that use procedural generation [19] can create large environment collections.

### 2.2 Procedural Generation in Reinforcement Learning Environments

The implementation of the procedural generation process determines the constraints on the environment collection's diversity and the open-endedness of the tasks. The eponymous Procgen benchmark [5] includes 16 different games with generated environments, enabling a controlled diversity. Minihack procedurally generates environments based on NetHack's pre-existing broad set of gridworld assets [20]. The Avalon benchmark consists of 20 tasks in 3D procedurally generated worlds [1]. Notably, these tasks can be composed. Malmo [13] creates an API for Minecraft to enable RL training in its procedurally generated 3D worlds, while MineRL [10] expands the API to leverage datasets of pre-existing content, an effort that Minedojo [8] considerably expands. A recent, closely related work is XLand-minigrid [17], a framework similar to our OpenXLand implementation, but constrained to the gridworld environments of minigrid. Another closely related work is Neural-MMO 2.0 [22], which includes a task generation system similar to that of [17] and Team et al. [23]. We also mention MemoryMaze [18], a framework which generates random 3D mazes to evaluate the long-term memory capabilities of agents, because we reimplement MemoryMaze with U3 to demonstrate the framework's extendability in Section 4.4.

### 2.3 The XLand Environment Spaces

The frameworks that we draw our inspiration from are the family of XLand-related works from Deepmind on open-ended learning. Team et al. [24] first presents XLand, a Unity-based system

to produce endless environments for open-ended learning. In XLand, agents must interact with procedurally generated environments to move the state toward a "goal state" represented by a set of conditions that depend on the environment. Alchemy [25] is another Unity-based RL suite, where the agent must discover latent causal structures between stones and potions to compose state transitions toward a goal state. Combining concepts found in both XLand and Alchemy, the Deepmind Team et al. [23] introduces XLand 2.0, which leverages both diverse world and compositional task generation to create rich environment dynamics. In particular, they use XLand 2.0 as a testbed to study RL foundation models for agent adaptation and find that, with correct the learning design choices, RL agents can adapt to new environments and tasks on a timescale similar to humans. Unfortunately, neither XLand nor XLand 2.0 are open-source. The machine learning community can neither build upon the work, nor validate their promising results.

## 3   Background

In this section, we go over the necessary concepts from the Unity-related work of Ward et al. [27] and Team et al. [24, 23]. Existing foundation models depend on large datasets, but agents trained for fast adaptation through meta-RL may provide an alternative path to general intelligence [23]. However, careful consideration is required when designing the environments for these agents. If the environment is too simple, the agent will learn a direct mapping from the environment state to action, destroying any chance of generalization [4]. If the environment is too complex, sparse rewards may prevent the agent from learning any patterns [28]. The approach favored by XLand 2.0 is to employ *procedural generation* to create complex environments from a set of simple rules.Once the agent learns these rules, it can compose them together to generalize what it has learned to new environments.

In Ward et al. [27], agents rely on sensors to provide observations, such as a pixel rendering of their line of sight or their current score. Agents use actuators to act in the environment, including moving, jumping and grabbing objects. We act on the assumption that both Team et al. [24] and Team et al. [23] use a similar system, and we include RL policies as components of agents.

Both XLand and XLand 2.0 procedurally generate the 3D *world* in which agents act, including the floor topology, manipulable objects and agent spawn locations. XLand 2.0 has a second procedural generation process that creates *production rules*, which act as a grammar for a subset of state transitions. Specifically, production rules are made of 1) *conditions* on agents and objects, and 2) *actions*, such as spawning or removing objects. To obtain rewards, agents must manipulate objects according to production rules to reach a *goal state*, which is a boolean function on the environment state. Although production rules in XLand do not allow conjunctions or disjunctions of multiple conditions, "predicate" refers to both the goal state and the production rule conditions. Together, the agents, the world, goal states and production rules for XLand 2.0 make up a task.

The environment in XLand is partially occluded, which forces the agent to learn to engage in exploration and planning [15]. In XLand's 3D world, this occlusion occurs when terrain tiles block the agent's line of sight. For production rules, this occlusion occurs because the rules active in each environment are different. Thus, compositional generalization alone is insufficient for the agent to solve the task. The agent must also learn across a number of *trials*, which are sequential repetitions of the same task, in order to uncover the hidden information in the environment. The results from XLand 2.0 [23] suggest that their agents learned to explore the environment in a reasoned manner that took into account the underlying structure of the environment. For example, the agent would systematically explore the world to find all production rule objects. It would also attempt to combine production rule pairs in a systematic way and remember pairs that reacted to form new objects. In this way, the agent was able to quickly learn the layout and rules of the environment, and adapt to the new task on the same timescale as human participants.

A foundation model of adaptive agents, trained on a sufficiently complex procedurally generated environment may be able to apply this type of systematic learning to meta-learn novel tasks even beyond its original training domain. XLand 2.0 showed promise of out of domain learning in an experiment in which the agent had to learn to push, and not pick up, an object. However, the extent of the adaptive agent's generalization capabilities can be studied further. U3 provides a potential test bed in which such an agent can be tested not only on different configurations of the XLand 2.0 environment, but also on novel environments with different rules, graphics and mechanics.

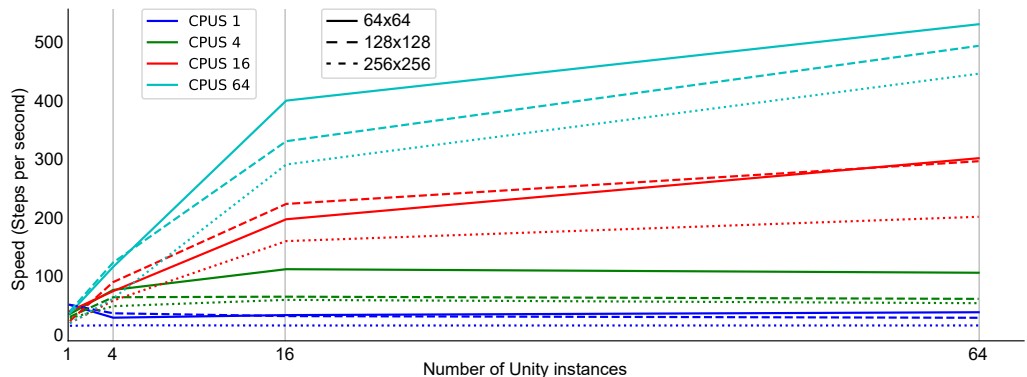

Figure 3: **Scalability.** We explore the scalability of U3 across 3 dimensions. The x-axis explores the effect of increasing the number of Unity instances running in parallel. The plot coloring indicates the number of CPUs available to the computation. Finally, the dotted vs dashed lines show the resolution of the visual observation.

# 4    United Unity Universe (U3)

U3 uses *procedural generation* to create a large number of reinforcement learning environments made of *worlds* and *production rules* in a fashion similar to XLand 2.0 [23]. A visual interface can control the active world and production rules, as well as their procedural generation processes. Both processes can also be extended with new components, enabling endlessly diverse environments.

## 4.1    Procedural generation in U3

We implement a *world generation process* that determines the static 3D floor plan. Inspired by the Wave Function Collapse (WFC) [9] and Model Synthesis [16] algorithms, the process samples environment components that are locally compatible with neighboring components until the world is complete. The world is made up of *voxels* that tile the entire world in 3 dimensions. The width, length and height of this voxel grid defines the size of the world. During the world procedural generation process each voxel is populated by a single *tile*, the base unit of the U3environment. In order to increase the amount of structure available to the procedural generation algorithm we do not place tiles directly. Instead, we define *metatiles* which consists of multiple tiles grouped together to form a pre-existing piece of the floor plan (e.g. "platform" or "ramp"). Each of the tile's six faces has a *type*, and we define legal adjacencies between face types such that, for any given step in the world generation process, the set of permissible metatiles that can be sampled is clearly defined. We also make it possible to dynamically reweight the probability distribution across permissible metatiles, e.g. to favor the sampling of specific environment features or to encourage metatile diversity within an environment.

Production rules are used to define a task on the generated world. We allow for statically defined rules, as well as a *production rule generation process* that controls the scaffold for how tasks are defined. We also offer an implementation to enable and disable rules during runtime, allowing for a huge variety of different task definitions. We explore the details of OpenXLand's production rule procedural generation in section 5

The combination of the world and production rule generation processes makes available a large, diverse set of environment dynamics. To facilitate access to these environments, U3 provides a direct visual interface to not only *control* the world, production rules, and visual aspects, but also *extend* the generation processes for the world and production rules. For example, users can modify the world generation process by changing the set of metatiles or the size of the world. They can also expand the production rule generation process with new conditions or actions. The fine-grained control and extendability of U3 result in an ideal ecosystem for scalable RL methods (See Section 4.4).

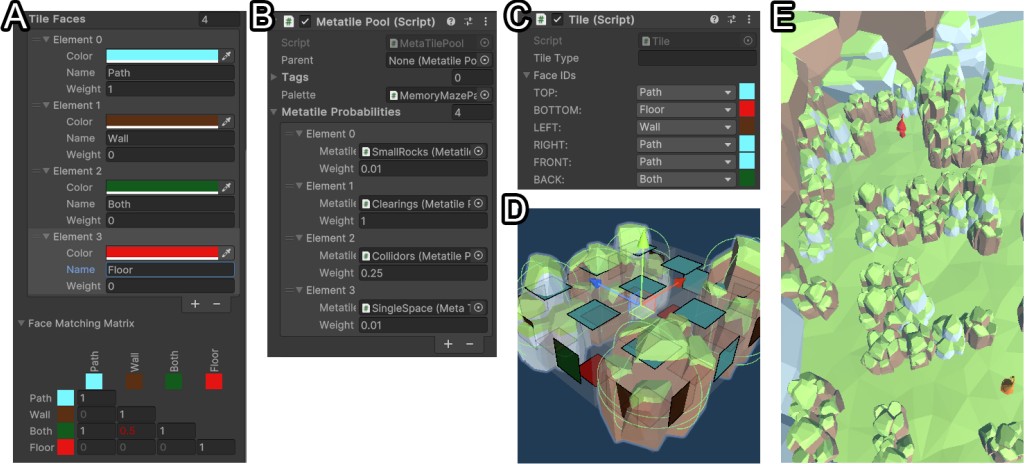

Figure 4: **Implementation of MemoryMaze.** A walk-through of the steps required to implement MemoryMaze in U3. **A)** First step is to define the tile face palette. You can assign a name and color to each face type for easy identification later. The face matching matrix defines what faces can be adjacent to each other, and how likely that adjacency should be. **B)** The next step is to define a weighted metatile pool. These pools are nestable, allowing for easy modification of weights across categories. **C) and D)** Next we set up the metatiles that populate the pool. Tiles type is used to calculate the permissible area of the generated terrain. Each face of the tile is assigned a face type. These tiles make up the structure of a metatile. **E)** Final result of the U3 set up for MemoryMaze. Note that the colored dots in the image are mushrooms that act as the goals.

## 4.2 U3 scales to multiple instances

In this section, we explore the scalability capabilities of our computational environment. The focus is on evaluating the performance in terms of the number of steps per second (SPS) that can be executed under varying configurations. Specifically, we assess the impact of the number of asynchronous environment instances, the number of CPU cores, and the image size of the observations.

Figure 3 illustrates the relationship between the number of asynchronous environment instances and the SPS. As expected, the SPS increases with the number of environments, but it saturates with a high number of environments. However, the saturation point increases with the number of cores per CPU, making the U3 scalable with the increase of compute capabilities. Finally, we tested the effect of the image observation size, specifically varying the size among 64x64, 128x128, and 256x256 pixels. As expected, increasing the image size results in a marginal decrease in the SPS.

## 4.3 U3 is easily extendable

Although our primary contribution is a U3 implementation of OpenXLand, U3 is also a powerful ecosystem for generating novel environments due to its extendability. Firstly, U3 can incorporate new metatiles with novel assets, be they acquired from the Unity Assets Store [14] or made by the user. We use only free assets on the Asset Store under the Asset Store's standard End User License Agreement. The world generation process can then be set up to construct novel environments with these new assets. The production rule generation process can also integrate new production rule conditions, actions and objects. Along with the fact that U3 enables production rules with conjunctions of conditions, this results in an infinitely rich space of possible production rules. As both the world and task generation processes are extendable, U3 can efficiently reimplement existing RL benchmarks.

We demonstrate the power and ease of use of U3's interface by implementing MemoryMaze [18]. Figure 4 details the process and user interfaces involved in the process of setting up the world generation. The task itself is set up by defining static production rules (one for each goal). The condition of these rules is AGENT_NEAR and the action is a custom defined action that is a composite of REWARD and TOGGLE_RULE. Note that by defining rules in terms of composites of existing

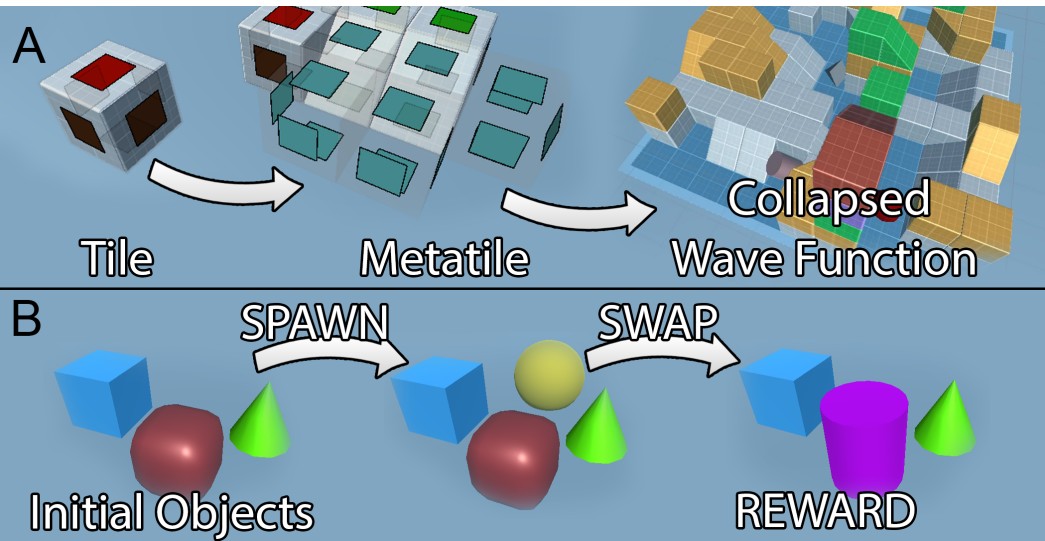

Figure 5: **Procedural generation in OpenXLand.** Illustration of the building blocks that make up U3's procedural generation system. **A)** The primary building block of the world is a tile. Tiles have faces that can be assigned face types. Metatiles allow the user to arrange tiles into distinct structures. Here we have illustrated one of the ramp metatiles used in OpenXLand. Metatiles also allow the user to define a payload, which is deposited into the scene after WFC. **B)** Production rules are generated based on a set of initial production rule objects. A production rule is created by sampling a condition and an action (from SPAWN or SWAP). The effect of these rules on the set of active production rule objects is calculated and the process is repeated. At the end of the chain, a final production rule is created by sampling a condition and applying the REWARD action.

rules U3 is able to create modular rules that can be reused by the community. Only one of the static rules is active at a time, and the active rule is communicated to the agent through a simple UI widget.

The U3 implementation of MemoryMaze (see C) uses free assets from the "Free Low Poly Nature Forest" asset pack under the Standard Unity Asset Store EULA.

### 4.4 U3 allows dynamic, adaptive control over the environment

U3 utilizes ML-Agents to provide a two-way communication channel between Unity and Python. Specifically, U3 provides a protocol for high-level manipulation of the environment by defining the parameters of the environment generation process, including the world generation process and production rule generation process. For example, it is possible to dynamically reweight the environment generation parameter distributions, enabling the possibility for agent-environment co-evolution and runtime adaptive auto-curricula. It also allows Unity to send meta-data about the state of the environment beyond the main training observations. This can be useful for designing repeatable experiments in the environment for evaluating trained agents.

## 5 OpenXLand

To demonstrate the utility of U3, we use it to implement OpenXLand, a task space similar to that of XLand [24] and XLand 2.0 [23]. We define XLand-style procedural generation processes for both environments and production rules. We also make available datasets of world and production rules, which can be used to build up to 36 trillion unique environments. In the simulated environment, the agent perceives a first-person visual representation of its surroundings. The agent is capable of executing both continuous and discrete actions. Continuous actions include moving forward or backward and rotating its body left or right, with the ability to modulate the velocity of each movement. Discrete actions encompass picking up objects, placing them down, TODO. The agent is positively rewarded with a reward of +1 for each time step during which the predefined goal condition is met. Each trial concludes after a user-specified number of steps, and the episode terminates

following a user-defined number of trials. To bootstrap the development of adaptive agents based on OpenXLand, we also provide clean implementations of common RL algorithms such as SAC, as well as a training system that supports repeated iterations and sequences of environments. Details on the metatiles, production objects, conditions, and actions used in OpenXLand can be found in the Appendix.

## 5.1 Datasets

In an effort to reduce the computational load of procedural generation and filtering at run-time, we provide a series of curated datasets for the U3 implementation of XLand. These datasets serve as a community benchmark suite that researchers can use to test and compare reinforcement learning algorithms and network architectures on a standardized set of XLand environments.

We provide 6 datasets of one million pre-generated worlds and 6 datasets of one million pre-generated production rules for a total of 36 trillion possible environment combinations. Curriculum learning has been shown to be important for task learning [2, 26]. Thus, we designed our datasets to facilitate the construction of a smooth learning curve, while still providing quantifiable increases in difficulty measures (Easy, Medium, and Hard). Datasets are drawn from a distribution of difficulties with some overlap between each progressive increase (See Figure 6). Information on the amount of time it takes to generate the datasets can be found in Table 5.

### 5.1.1 World datasets

The difficulty of the world datasets is given by two measures: world size, and world height. World size is the primary difficulty metric that defines the Easy, Medium, and Hard datasets. Each world was constructed by drawing a world width and length from a Gaussian distribution with parameters defined by the dataset. We set the minimum size of the world to be 5x5, so that all dataset distributions were truncated at that minimum bound. Details on the parameters used for each dataset can be found in Table 3 in the Appendix.

World height serves to modulate the topological complexity of the worlds. Low heights lead to better visibility, more simply connected terrain, and larger accessible areas. We provide two types of world heights for each world size: Low and High. World height cannot be less than 1, so all distributions are truncated appropriately. World height provides an orthogonal measure of complexity that can be used to further enhance the curriculum of training.

### 5.1.2 Production rule datasets

Production rule datasets also include two measures of difficulty: chain length and distractor count. Chain length is the number of SWAP or SPAWN rules that must be activated to reach the REWARD rule. Similarly to the world datasets we have drawn chain lengths from Gaussian distributions with parameters defined by the dataset difficulty. We also draw the number of initial objects from the same Gaussian distribution as the chain length. We lock chain length and number of initial objects to the same distribution to simplify the complexity of the datasets. All chains terminate in a REWARD rule, so the minimum chain length is 1. Similarly, there must be at least 1 initial object, so all distributions for chain length and initial objects were truncated at 1. Details can be found in Table 4 in the Appendix and in Figure 6.

To ensure a consistent set of production rules that allow for a valid solution, we first sample production rules with mutually exclusive conditions, such that there is always a valid path to the reward. We subsequently sample "distractors" rules that branch off from the main production rule. Distractor count adds additional complexity by creating rules that act as distractors or potentially cause the agent to become unable to achieve the reward. A distractor becomes a dead end when the production rule removes an object from the environment such that the production rule chain can no longer be completed. Thus, the severity of these distractors can be dependent on the progress of the agent through the chain. We provide two levels of distractor counts: Low and High. We truncate all distributions at a minimum value of 0 distractors.

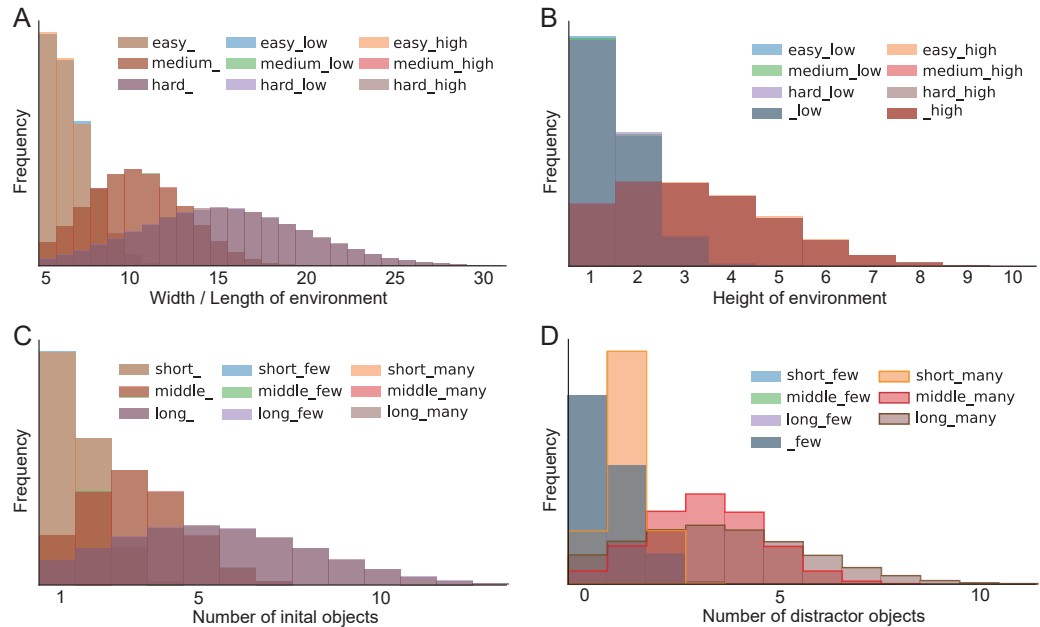

Figure 6: **Dataset distributions.** Histograms showing the distributions of key dataset metrics. Note that all datasets have overlap. **A)** Distribution of width and lengths in the world datasets. Note that the 6 datasets are split into "easy", "medium" and "hard" for this metric. **B)** Distribution of heights in the world datasets. Note that now the 6 datasets are split into "low" and "high". **C)** Distribution of the number of initial objects in the rule datasets. The 6 datasets are split into "short", "middle" and "long". **D)** Distribution of the number of distractors in the rule datasets. Note that the "few" datasets collapse to the same distribution. The "many" datasets all have unique distributions. We have outlined the "many" datasets to make their overlap more clear.

### 5.1.3 Dataset deployment

All datasets are hosted on Huggingface for easy download and usage. Datasets are distributed as a tar file of 1 million JSON files. The individual JSON files are labeled from 1 to 1 million, allowing easy splitting of the dataset into train and test partitions. While the generation of novel environments requires the user to download the Unity Editor and work directly with U3's built-in UI and classes, we provide Python scripts that allow the generation of new OpenXLand (or any existing environment) datasets without the need to install the Unity Editor.

## 5.2 Evaluation protocol

To standardize future performance comparisons on U3 and OpenXLand, we propose an evaluation protocol that assesses the agent's ability to generalize to out-of-distribution tasks. We designate a held-out set of production rule objects. Then, we filter each dataset for instances that contain at least one of those elements. The remaining instances form the training set, while the testing set contains the excluded elements. Therefore, the training split does not contain any instances with these specific objects or conditions, ensuring that the rules and goals encountered during the testing phase are entirely novel.

Following the methodology of previous works [23, 17], we evaluate the agent using the 20th percentile of the normalized return for each environment instance, as this avoids simpler tasks dominating the average performance of the agent. Furthermore, we propose restricting the method comparison to within datasets, as opposed to across datasets, as this avoids the costly normalization by compute-heavy fine-tuned baselines proposed in [23].

### 5.3 Results

We tested our environment using the Soft Actor-Critic (SAC) implementation available in CleanRL[12]. Each episode consists of 8 trials, each having 481 steps, corresponding to 40 human seconds. The procedurally generated world is loaded from the easy_low dataset at the start of an episode and kept constant across all 8 trials. The production rules are loaded from the short_few dataset. The initial positions of the production rules and the agent are randomized to locations in the largest accessible area of the world at reset at the beginning of each trial. The agent's context is kept across trials but reset when the episode ends. Results of the training are available in the Appendix.

## 6 Limitations

One of the main obstacles to using Unity for RL is correctly setting up Unity. Despite U3 including all relevant software files, hardware and platform differences can complicate installation. In the future, we plan to implement a docker environment to ease installation considerations. Another obstacle to using Unity as an RL environment can be the compute required to train an agent in Unity.[2] We expand on the efficiency of the current implementation of U3 in Appendix F. We also note that an important line of future work involves optimizing the efficiency of the current implementation of U3 to make it more accessible, such as environments like [17] and [21, 22].

This initial release of U3 and OpenXLand is missing some features. Despite the current implementation of production rules generalizing Xland 2.0, certain conditions are missing (e.g. "seeing" for agents and objects). Our implementation is also missing the ability to pass information about the active production rules to the agent. Another feature we plan to implement in future work is the ability to test human participants on U3 tasks using a web interface (This functionality was present in an early version of U3 [3]). Finally, our training scripts currently lack teacher models and auto-curricula. Future work should also include implementing features beyond the original XLand 2.0. Production rules are a powerful system that we can generalize far beyond the simple chains of XLand 2.0. Although the current production rule system already goes beyond that of XLand 2.0 with conjunctions of conditions, enabling even more generality would require updating the implementation.

## 7 Conclusion

We present U3, an open-source ecosystem capable of implementing and defining procedurally generated 3D environments with a simple user interface. U3 is scalable to many instances and extendable with new assets for the generation of both environments and production rules. We also provide OpenXLand, an open-source implementation in the style of XLand 2.0 [23], an open-ended environment that has shown promise in training adaptive agents. As the science of foundation models progresses, the adaptation capacities of AI agents in novel and open-ended environments stand out as a key yardstick of general intelligence. We hope that U3 will facilitate the training and evaluation of such foundation models in the open-source community.

## Broader Impact

As far as we know, Unity has yet to solve intelligence [27]. However, XLand 2.0 [23] suggests a promising new approach to general intelligence with adaptive agents that are able to actively explore and meta-learn their environment. By improving the adaptive ability of agents, we can reduce the risk of unintended consequences from agents in unfamiliar settings. We hope that U3 will bolster the study of adaptive agents by the open-source community, and facilitate research on the strengths, limitations and human correlates of such foundational models. An important limitation of U3 is its accessibility due to the compute requirements of training agents such as Ada. Our commitment to open-source ensures that any work stemming from the U3 ecosystem will be transparently assessable against ethical guidelines, ensure robustness across many distinct tasks, be reproducible across research groups and available to all researchers across the globe. However, since the code is open source and users can create any assets they see fit, we have limited control and visibility over the downstream uses of U3.

---

[2]Training an agent to completion in [23] took over 50000 TPU3 device hours.

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

# A  Acknowledgments

We acknowledge the support from the Canada CIFAR AI Chair Program and from the Canada Excellence Research Chairs Program. This research was made possible thanks to the computing resources on the Summit and Frontier supercomputers, provided by the Oak Ridge Leadership Computing Facility at the Oak Ridge National Laboratory, which is supported by the Office of Science of the U.S. Department of Energy under Contract No. DE-AC05-00OR22725.

# B  OpenXLand Components

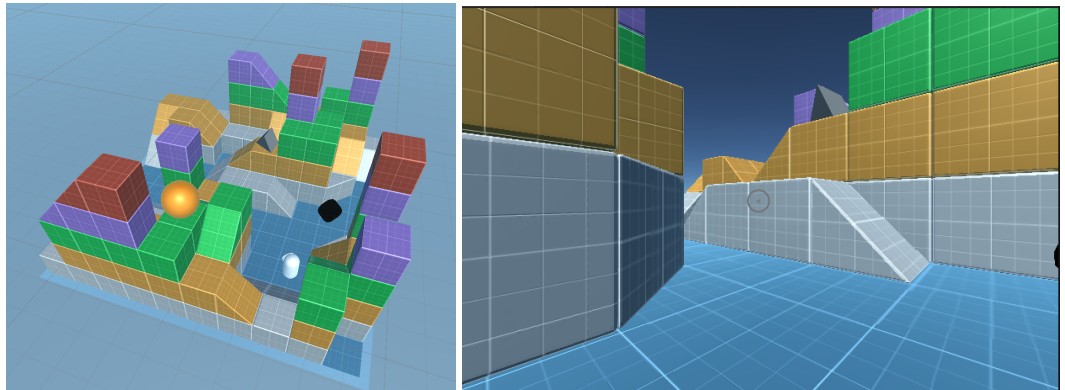

Figure 7: **Left:** An example OpenXLand environment. In this environment, the agent receives reward when the orange sphere makes contact with the blue pyramid. We see that the orange sphere is elevated, and therefore the agent must find it and use the ramps to access it. As to the blue pyramid, we do not see it because it is not there: the agent must first get the orange sphere near the black rounded cube first to spawn one. This environment also contains a grey pyramid that serves as a distraction. Importantly, if the agent brings the grey pyramid near the black rounded cube, both will disappear, making it impossible for the agent to spawn a blue pyramid and subsequently obtain its reward. **Right:** The agent's perspective at initialization. The agent does not see the orange sphere, and so will have to search for it. However, the agent does see the grey pyramid, which could act as a distraction. As pointed out in [23], solving tasks in OpenXLand requires many skill such as navigation, exploration, experimentation, and avoiding irreversible transitions. However, since the innumerable tasks are composed of a small number of components, the agent can learn to reuse previously acquired knowledge to improve performance.

OpenXLand is an open-source implementation of the environment frameworks known as XLand [24] and XLand 2.0 [23]. OpenXLand generates 3D worlds from building blocks known as *metatiles*. In the world, agents are expected to solve tasks defined by *production rules*, which trigger based on the state of the agent and objects in the environment. Figure 7 shows an example of such a world and task.

Below, we go over specific details of the metatiles and the production rules.

## B.1  Metatiles

The OpenXLand framework uses procedural generation to create diverse environments from simple building blocks we call *metatiles*. During generation, the environment is filled with metatiles. These metatiles are designed to create interesting multi-level floor topologies: agents can use ramps to access platforms (see Figure 8). Metatiles for OpenXLand can be found in the Resources/GameObjects/XLand/MetaTiles subfolder of the Unity asset folder. There are 13 total metatiles distributed across two pools: a medium pool with a weight of 10, and a small pool with a weight of 1. This means that the procedural generation algorithm will attempt to place medium metatiles much more frequently than small metatiles. A full list of metatiles and their relative weights can be found in Table 1 However, the small metatiles still serve an important role as they can fill in gaps where medium metatiles no longer fit. Note that AirHelper is meant to be used only when no other tiles are available.

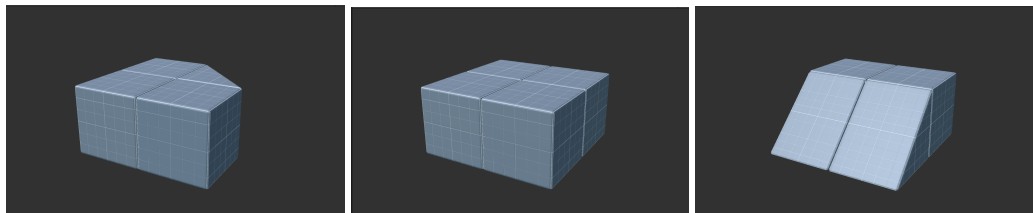

Figure 8: Three examples of metatiles. On the left, we see a platform with a rounded corner. The middle contains a square platform. The right contains a ramp, which the agent can use to traverse multi-level floor topologies.

Table 1: Metatiles used in OpenXLand. The sampling weight is used to calculate how likely a tile is to be sampled. U3 makes it possible to easily adjust these weights to enrich environment generation dynamics.

| Name | Description | Sampling Weight |
|---|---|---|
| RampInset | 2 tile wide ramp with tiles on both sides | 1500 |
| Ramp1Edge | 2 tile wide ramp with tiles on one side | 750 |
| RampNoEdge | 2 tile wide ramp with no tiles the sides | 750 |
| Corner | 4 tile platform with one corner rounded | 1000 |
| PlatformNoEdge | 4 tile platform with no internal edges | 100 |
| Platform1Edge | 4 tile platform with a single internal edge | 100 |
| Platform2Edge | 4 tile platform with a two internal edges at the corner | 100 |
| Platform4Edge | 4 tile platform with a four internal edges | 100 |
| AirPocket | 4 empty tiles | 100 |
| SmallRamp | A minimal 1 tile wide ramp | 450 |
| Air | A empty tile | 45 |
| WhiteTile | A single tile | 4 |
| AirHelper | A tile that allows an empty tile to connect to internal edges | 0.0004 |

### B.1.1 The Face Matching Matrix

The logic for how metatiles fit together is defined by the face matching matrix. This adjacency matrix allows users to define which faces can be placed next to other faces, and with what weight. U3 will attempt to place metatiles at frequencies that respect both the face matching matrix frequencies and the metatile pool weights, however, due to the complexity of the generation process there is a fair amount of variance in the actual number of each metatile and face-face pairs placed.

OpenXLand uses 7 face types. Bedrock denotes the bottom of tiles, so Air and ramps cannot be placed below it. Similarly, Surface denotes the top of a tile and can match with any type of face, but attempts to avoid Air (0.05 weight). Air and Ground are generic face types of empty and solid faces respectively. InsideEdge attempts to connect to other InsideEdge faces, to create larger platforms. RampTop and RampBottom attempt to connect in a mutually exclusive manner to make sure that multiple ramps can chain together in the correct way.

### B.1.2 Wave Function Collapse Algorithm

U3 implements a 3D wave function collapse algorithm with a several optimizations. The original wave function collapse algorithm tracks all possible configurations of each world voxel to calculate the entropy of the voxels. We simplify this calculation by only considering whether a given voxel is [Andrew: ~~down~~ restricted] to a single permissible metatile configuration. If this is the case, that voxel is chosen [Andrew: to be placed next] as the next location to place a metatile. If no voxel has only 1 remaining configuration, then the next voxel is chosen randomly from the current wavefront.

### B.2 Production Rules

A key element of XLand 2.0 is *production rules*, which define compositional tasks that agents are meant to complete. Agents 1) manipulate production rule objects 2) to satisfy production rule

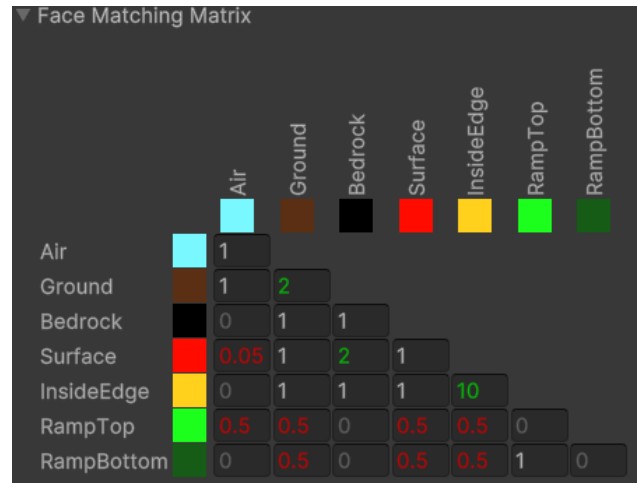

**Face Types**

| Face Types |
| --- |
| Air |
| Ground |
| Bedrock |
| Surface |
| InsideEdge |
| RampTop |
| RampBottom |

Figure 9: **Left:** The face matching matrix between face types in OpenXLand. **Right:** The face types in the adjacency matrix. Each metatile is made of tiles, which have faces, and the adjacency matrix determines which faces can be neighbors. The adjacency matrix constrains the set of permissible metatiles that can be placed at a given location during the environment generation process.

conditions, which 3) trigger production rule actions that transition the state. The goal of the agent is to trigger these production rules until the environment reaches a goal state and the agent receives reward. In Table 2, we outline the production rule objects by shape and color, as well as the production conditions and the production rule actions. Naturally, each set can be extended.

Table 2: A table containing the main components of OpenXLand's production rule system. Our set of objects, which is specified by shape and color, is similar in nature to that of XLand 2.0 [23], though with 7 objects and over 20 colors. As for the conditions We extend those in Team et al. [23] with the new conditions "drop", "pickup" and "throw". Also, whereas in XLand 2.0, objects that are part of production rule conditions disappear when the condition is fulfilled, we include the possibility of objects remaining in the environment. The "spawn" action creates a new object, the "swap" action swaps the production rule condition's subject, object or both for a new object, the "remove" action simply removes an item from the environment, and the "reward" action gives reward to the agent. Therefore, our production rules go beyond forcibly removing the condition's objects or swapping them for new ones.

| Object Shapes | Object Colors | Conditions | Actions |
| --- | --- | --- | --- |
| Cube | red | NEAR | SPAWN |
| Sphere | green | CONTACT | SWAP |
| Pyramid | blue | HOLD | REMOVE |
| Cylinder | purple | PICKUP | REWARD |
| Cone | yellow | THROW | |
| Rounded Box | grey | DROP | |
| Rounded Rectangle | white | | |
| | etc. | | |

### B.2.1 Production Rule Objects

Our production rule objects are quite similar to those of Team et al. [23]: as us, they use cubes, spheres and pyramids (not counting walls), but we also include other objects such as cylinders, cones, and rounded boxes and rectangles. Team et al. [23] uses three colors: black, purple and yellow. We

see no reason a priori why there shouldn't be more colors, so we include over 20 colors, a few of which can be found in 2. This raises the unique number of object-shape pairs from 9 to over 150. Naturally, we can restrict or extend the set of colors. Furthermore, although OpenXLand's objects are identified with simple colors and shapes, it is possible to use U3 to identify objects with different characteristics.

### B.2.2 Production Rule Conditions

We implement the following conditions from Team et al. [23]: "near", "contact" and "hold". We plan on implementing the final condition, "see" in future work. On top of this, we implement three new conditions: "pickup", "throw" and "drop", which we implement as callback functions that trigger based on the agent's behaviour. The agent can only hold (and therefore pickup, throw and drop) one object at a time.

### B.2.3 Production Rule Actions

In Team et al. [23], production rules and goal states are represented as separate concepts. We find it more amenable to represent them as instances of the same concept by turning goal states into production rules that reward the agent as an action. This allows us to represent production rules and goal states as part of the same language. We also include in our production rule actions the possibility of preserving the objects present in the production rule conditions, which can allow for distinct production rules that repeatedly depend on the same object.

## C   Implementing MemoryMaze: A Walkthrough

MemoryMaze [18] differs from OpenXLand in a few keys ways: First, the world is flat, so we only need a height of 1. Second, production rules are not procedurally generated, but are instead attached to "goal" objects spawned throughout the world. Third, only 1 "goal" is active at a time. Fourth, there is a UI element that displays the current "goal".

To illustrate the ability of U3 to render more naturalistic worlds, we use a low polygon forest pack from the Unity asset store. Thus, we modify the original MemoryMaze by replacing the walls with rocks, and the goals with forest mushrooms of various colors.

Our MemoryMaze starts by procedurally generating the flat world using the same metatile system as OpenXLand. We want goals to be spread out and only spawn in small clearings. Thus, we define a new class (ProductionRuleSpawn) that defines locations where goals can spawn, and add it to the Clearing metatile itself. By adding a script into the metatile we are able to use the existing procedural generation system to add structured randomness to this new spawning logic. We also include custom spawn logic in the script to spawn only up to 5 goals, and make sure that each goal is a minimum distance from the previously spawned goals.

Further, we add logic to ProductionRuleSpawn such that a new production rule is created for each spawned goal. These rules all use a new NEAR condition that triggers when the agent is near the production rule's subject (in this case, the goal object). Because of the modular design of U3, this new condition can now be used in any newly generated OpenXLand environments as well. We also create a new REWARD_TOGGLE action that is a compound action of a REWARD plus new logic that deactivates the current production rule, and activates a different production rule at random. To implement this logic we also add a new flag to production rules that allows them to be turned on and off at run-time. In this way, as new environments are added to the U3 ecosystem, we expect that new and more powerful features will also be added to the base OpenXLand environments.

The final missing piece of the MemoryMaze environment is the UI element that shows the current goal. We adopt the same UI as the original with a colored border around the image. Unity is a powerful 3D game engine with native support for 3D rendering, UI, sprites, materials, etc, so creating this UI took only 1 function call into Unity's API (GUI.DrawTexture).

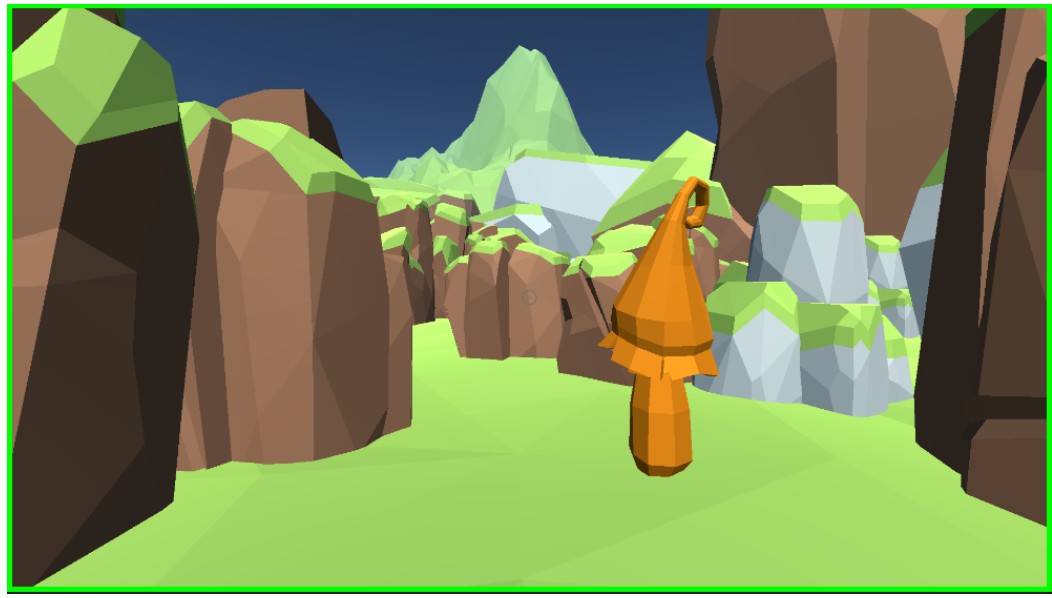

Figure 10: MemoryMaze environment observation. Our version of MemoryMaze uses naturalistic assets. We hope that harnessing the variety of assets available will help to increase agents' overall adaptability.

## D   Dataset Details

In this section, we go over the details of the datasets. We split the datasets into 6 world datasets (see Table 3), and 6 production rule datasets (see Table 4). Each dataset contains 1 million environments (see Table 5 for generation times). We design the datasets with overlapping parameters to facilitate smooth curriculum learning. We filter the world datasets to include only those worlds that have a largest accessible area of over 50% of the total number of tiles in the X-Z plane.

We find the largest accessible area by first constructing a local, directed connectivity graph using the specific configuration of each voxel. For simplicity, we consider only the cardinal connections between voxels, and only connections in the width-length plane (we ignore connections within a single column of voxels). Two voxels are connected bi-directionally if they are at the same terrain height and higher voxels are connected unidirectionally to lower tiles - representing the agent's ability to fall to the lower level. Ramps serve as the sole method for connections from a lower layer to the next highest layer. Once the connectivity graph is constructed, the largest accessible area is found by computing the largest strongly connected sub-component.

Distractors' positions in the production rule chains heavily influence their impact. For example, if the agent has already completed the step that requires a Yellow Cube, then removing the Yellow Cube is merely a distraction. However, if the Yellow Cube is still required, then its removal creates a dead end. Due to this complexity, we do not directly consider distractors vs dead ends in our difficulty measure, but note that as more distractors are added the probability of dead ends also increases.

## E   Training Results

To demonstrate the learnability of our environment, we randomly select a task from the easy dataset and applied the Soft Actor-Critic (SAC) [11] method. Image observations are processed using a convolutional neural network for both the actor and Q networks, each comprising approximately 2 million parameters. The SAC agent is trained for 75 million environment steps, utilizing a replay buffer that stored 1 million transitions. While these parameter choices may appear modest compared to those in the Ada paper [23] and may not fully exploit emergent meta-learning capabilities, our primary aim is to underscore the learnability of our environment, not to recreate their agent.

Table 3: World Datasets. These values were used to define Gaussian distributions used to sample parameters during dataset construction. We note that the harder we consider environments to be, the bigger they are.

| Name | Width and Length | | Height | |
|---|---|---|---|---|
| | $\bar{x}$ | $\sigma$ | $\bar{x}$ | $\sigma$ |
| easy_low | 5 | 2 | 1 | 1 |
| easy_high | 5 | 2 | 3 | 2 |
| medium_low | 10 | 3 | 1 | 1 |
| medium_high | 10 | 3 | 3 | 2 |
| hard_low | 15 | 5 | 1 | 1 |
| hard_high | 15 | 5 | 3 | 2 |

Table 4: Production Rule Datasets. These values were used to define the Gaussian distribution used to sample parameters during dataset construction. Since we sample the production rule chain lengths from distributions, the datasets of shorter chains will still contain long chains, although the dataset will be dominated by shorter, less complicated tasks. As the average length of chains increases throughout the datasets, the number of distractors also increases.

| Name | Chain Length and Initial Objects | | Distractors | |
|---|---|---|---|---|
| | $\bar{x}$ | $\sigma$ | $\bar{x}$ | $\sigma$ |
| short_few | 1 | 1 | 0 | 1 |
| short_many | 1 | 1 | 1 | 0.5 |
| middle_few | 3 | 1.5 | 0 | 1 |
| middle_many | 3 | 1.5 | 2 | 1.5 |
| long_few | 5 | 3 | 0 | 1 |
| long_many | 5 | 3 | 3 | 2.5 |

Table 5: Average time to generate an Environment in OpenXLand, by environment difficulty. Recall that environment difficulty impacts the width, length and height of the environment (see Table 3). To generate the datasets of 1 million environments, the Easy datasets took approximately 10 minutes per dataset, the Medium, 2 hours, and the Hard, 2 days.

| Difficulty | Average Environment Generation Time (ms) |
|---|---|
| Easy | 0.6 |
| Medium | 7.2 |
| Hard | 172.8 |

Figure 11 illustrates that the episodic return of our learning agent increases throughout training, surpassing a random agent baseline's performance. We also extend our evaluation to the entire easy dataset, conducting 8 trials per episode on different tasks. Interestingly, our learning agent exhibits minimal improvement over random agents in this broader context, indicating the limitations of conventional methods. We observe similar outcomes when applying the method to medium datasets, where random agents fail to accrue any rewards, underscoring the challenge of reward collection within these datasets.

# F   U3's efficiency

# G   Resources

Most of the resources for this project were spent on implementing the baselines. The CPU compute spent to generate the envs amounts to 200 hours. The GPU compute spent to train the baselines amounts to around 140 hours of NVIDIA A100-SXM4-80GB.

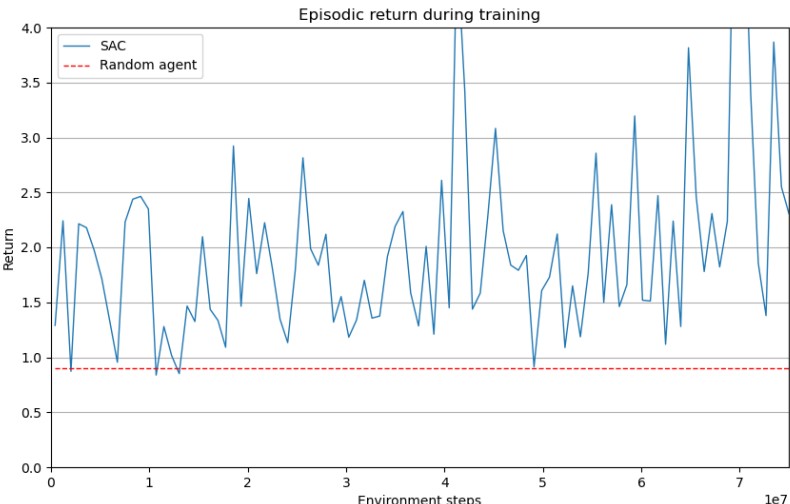

Figure 11: Episodic return during single environment training of the SAC agent compared to a random agent. We train the SAC agent for 75 million environment steps. This required approximately 48 hours on a single A100 GPU, a small fraction of the 53760 TPUv3 device hours it took to train Ada [23]. The episodic return for the SAC agent shows significant fluctuations but generally demonstrates a higher return compared to the random agent baseline, represented by the red dashed line.

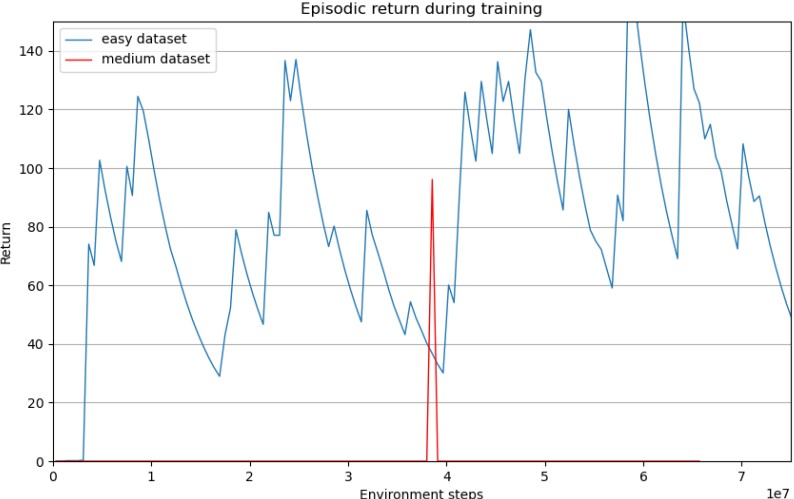

Figure 12: Episodic return during training using SAC on the whole dataset with 8 trials per episode and changing the task each episode. Collecting reward on the medium dataset is hard showing the limitations of the conventional methods. Collecting reward on easy dataset is easier, but meta-learning requires more advanced methodologies.

| Env size | Prod objs | Prod rules | Script time (ms) | $\sigma$ |
|---|---|---|---|---|
| 10x105 | 3 | 2 | 3.04 | 3.23 |
| 100x10x5 | 3 | 2 | 4.76 | 4.14 |
| 200x10x5 | 3 | 2 | 6.77 | 5.03 |
| 10x10x5 | 30 | 2 | 4.24 | 3.44 |
| 10x10x5 | 300 | 2 | 65.29 | 5.31 |
| 10x10x5 | 3 | 20 | 3.5 | 3.69 |
| 10x10x5 | 3 | 200 | 5.42 | 2.68 |
| 10x10x5 | 3 | 2000 | 10.44 | 4.24 |

Table 6: **Performance scaling of Unity environment.** We explore the effect of environment complexity on U3's performance. Note that due to time constraints we compare the runtime of the U3 scripts within the Unity editor – not the true SPS of the headless Linux build. Note that the script runtime of the 10x10x5 environment is ∼3ms, or 300 SPS. Yet, the SPS of a single environment was ∼31, or 30ms per step, which suggests that the U3 script runtime makes up only around 10% of the actual computation time. We test the complexity across three different axes: Environment size (Env size), Number of production rule objects (Prod objs) and Number of production rules (Prod rules). We vary each of these axes over ∼2 orders of magnitude and note that the only time the runtime reaches over 30ms is in the 300 object environment. Note that the environment generation time does scale super-linearly so that environments larger than 200x10x5 take prohibitively long to generate.

# H   URL to Dataset and Metadata

**Datasets**: `https://huggingface.co/datasets/cerc-aai/u3_datasets`

**U3 source code**: `https://github.com/CERC-AAI/u3/`

**Croissant Metadata**: `https://huggingface.co/api/datasets/cerc-aai/u3_datasets/croissant`

We plan to fully open-source the code for U3, including the Unity environment framework, the python training code and the pipeline. Using U3 still requires installing Unity, which impinges on reproducibility. As stated in Section 7, we plan on containerizing the environment to democratize access to U3.

# I Datasheet

## I.1 Motivation

1. **For what purpose was the dataset created?** *U3 and its associated OpenXLand datasets were created to promote the open development of foundation models for RL and meta-RL.*

2. **Who created the dataset and on behalf of which entity?** *U3 and its datasets were created by the researchers listed in the author list.*

3. **Who funded the creation of the dataset?** *The main funding bodies include the Canada Excellence Research Chairs Program, as well as the resources of the INCITE program award "Scalable Foundation Models for Transferable Generalist AI" provided by Oak Ridge Leadership Computing Facility at the Oak Ridge National Laboratory, which is supported by the Office of Science of the U.S. Department of Energy under Contract No. DE-AC05-00OR22725*

## I.2 Distribution

1. **Will the dataset be distributed to third parties outside of the entity (e.g., company, institution, organization) on behalf of which the dataset was created?** *The U3 framework and its datasets will be publicly available.*

2. **How will the dataset be distributed (e.g., tarball on website, API, GitHub)?** *The datasets will be distributed through Hugging Face, while the code will be made available on Github.*

3. **Have any third parties imposed IP-based or other restrictions on the data associated with the instances?** *No.*

4. **Do any export controls or other regulatory restrictions apply to the dataset or to individual instances?** *No.*

## I.3 Maintenance

1. **Who will be supporting/hosting/maintaining the dataset?** *The CERC-AAI Lab at Université de Montréal will maintain the U3 framework and its datasets.*

2. **How can the owner/curator/manager of the dataset be contacted (e.g., email address)?** *The managers of the dataset can be contacted at connor.brennan@mila.quebec, andrew.williams@mila.quebec and irina.rish@gmail.com*

3. **Is there an erratum?** *No. We will release one in the future if necessary.*

4. **Will the dataset be updated (e.g., to correct labeling errors, add new instances, delete instances)?** *Yes, the datasets will be updated if necessary.*

5. **If the dataset relates to people, are there applicable limits on the retention of the data associated with the instances (e.g., were the individuals in question told that their data would be retained for a fixed period of time and then deleted?)** N/A

6. **Will older versions of the dataset continue to be supported/hosted/maintained?** N/A

7. **If others want to extend/augment/build on/contribute to the dataset, is there a mechanism for them to do so?** *U3 welcomes contributors on GitHub. Should the need arise to extend the datasets on Hugging Face, we will establish a procedure for doing so.*

## I.4 Composition

1. What do the instances that comprise the dataset represent **(e.g., documents, photos, people, countries?)** *The instances that comprise the datasets are JSON datafiles that describe the necessary information to instantiate RL environments in Unity.*

2. **How many instances are there in total (of each type, if appropriate)?** *Six files of one million environment instances, six files of one million production rule instances, for a total of twelve million instances spread across 12 files.*

3. **Does the dataset contain all possible instances or is it a sample of instances from a larger set?** *Is it a subset of the effectively infinite number of instances that U3 can generate.*

4. **Is there a label or target associated with each instance?** *Each instance contains the label information necessary to instantiate a Unity environment.*

5. **Is any information missing from individual instances?** *No.*

6. **Are there recommended data splits (e.g., training, development/validation, testing)?** *No.*

7. **Are there any errors, sources of noise, or redundancies in the dataset?** *The instances are sampled with replacement from a procedural generation process, so yes there can be redundancies.*

8. **Is the dataset self-contained, or does it link to or otherwise rely on external resources (e.g., websites, tweets, other datasets)?** *The datasets rely on access to a Unity instance*

9. **Does the dataset contain data that might be considered confidential?** *No.*

10. **Does the dataset contain data that, if viewed directly, might be offensive, insulting, threatening, or might otherwise cause anxiety?** *No.*

### I.5 Collection Process

1. **How was the data associated with each instance acquired?** *We procedurally generated environments in Unity and collected the necessary information to recreate them using the U3 framework.*

2. **What mechanisms or procedures were used to collect the data (e.g., hardware apparatus or sensor, manual human curation, software program, software API)?** *We used the Unity Engine and Python, as well as GPUs for training RL baselines.*

3. **Who was involved in the data collection process (e.g., students, crowdworkers, contractors) and how were they compensated (e.g., how much were crowdworkers paid)?** *Students, paid through the funding sources described above.*

4. **Does the dataset relate to people?** *No.*

5. **Did you collect the data from the individuals in question directly, or obtain it via third parties or other sources (e.g., websites)?** N/A

### I.6 Uses

1. **Has the dataset been used for any tasks already?** *No, the data has not been used, other than for running baselines.*

2. **What (other) tasks could the dataset be used for?** *The datasets could be used for RL-related tasks as standalone or as part of a blended dataset*

3. **Is there anything about the composition of the dataset or the way it was collected and preprocessed/cleaned/labeled that might impact future uses?** *The use of U3 and its datasets relies on access to the Unity Engine.*

4. **Are there tasks for which the dataset should not be used?** *No.*

## J Author statement

We bear all responsibility in case of violation of rights.

We use the MIT License for our data.

