# OpenReview forum: "Using Unity to Help Solve Reinforcement Learning"
_NeurIPS.cc/2024/Datasets_and_Benchmarks_Track — NeurIPS 2024 Track Datasets and Benchmarks Poster_

### Official Review · Reviewer_Trj1 · 2024-06-25
**Using Unity to Help Solve Reinforcement Learning**

**Rating:** 6
**Confidence:** 4

**Review:**

The authors effectively demonstrate how U3's procedural generation capabilities can create a diverse range of RL environments, which are crucial for developing adaptive and generalizable learning systems.The scalability and extendability of U3 are well-illustrated through comprehensive explanations and examples, such as the reimplementation of the MemoryMaze benchmark. The inclusion of pre-generated datasets and RL baselines further enhances the practical utility of U3, making it a valuable resource for the RL community.Overall, this paper provides a well-rounded and impactful contribution to RL research, offering a versatile and scalable tool that could significantly accelerate advancements in the field.

**Strengths:**

U3 offers extensive control over environment and task generation, allowing users to modify terrains, task rules, and other parameters easily. This flexibility is essential for developing and testing adaptive and generalizable RL agents.
The paper demonstrates U3's scalability across multiple CPU instances, highlighting its potential for efficient training and experimentation. This makes it suitable for large-scale RL research and applications.
The toolkit includes pre-generated terrains, tasks, and RL baselines, which facilitate quick experimentation and benchmarking. This comprehensive set of resources is valuable for researchers looking to test new RL algorithms and models.

**Additional Feedback:**

1. Although the paper provides a good overview of U3's features, it could benefit from a deeper technical discussion on the underlying algorithms and design choices, particularly in the procedural generation processes and their impact on RL training.

2. The paper would benefit from a more detailed description of the reward composition mechanisms, which are crucial for understanding how different behaviors are incentivized within the reinforcement learning environments created using the U3 toolkit.

3. Does the paper address whether the consistency of the production rules in the U3 toolkit is checked, and if so, how is this consistency ensured?

4. Does the paper discuss whether the sequence of production rules is verified to ensure that the tasks in the procedurally generated environments are solvable?

**Clarity:**

The paper present complex concepts and technical details in an accessible and comprehensible manner, making it easy for readers to understand the significance and applications of the United Unity Universe (U3) toolkit.

**Correctness:**

The paper is well-written, clearly articulating the innovative features of the United Unity Universe (U3) toolkit and its potential impact on the field of reinforcement learning, while also acknowledging current limitations and outlining plans for future enhancements.

**Documentation:**

The supplementary material is well written.

**Ethics:**

There are not any ethical issues.

**Limitations:**

The Limitations section is well written although there are some weak points of this approach missing.

**Opportunities For Improvement:**

The paper lacks detailed performance comparisons between U3 and other existing RL frameworks. Including such comparisons would provide a clearer picture of U3's advantages and limitations relative to current tools. While the paper discusses the general benefits of U3, it does not delve deeply into potential limitations or challenges in specific use cases. For instance, the scalability of U3 with very high-dimensional observation spaces or its performance with extremely complex tasks could be explored further.

**Relation To Prior Work:**

The United Unity Universe (U3) toolkit extends the capabilities of the closed-source XLand project by providing an open-source platform that leverages Unity for creating diverse and customizable reinforcement learning environments.

**Summary And Contributions:**

The paper introduces the United Unity Universe (U3), an open-source toolkit designed to facilitate the creation of diverse reinforcement learning (RL) environments. Leveraging the Unity game engine's capabilities, U3 integrates procedural generation to produce a vast array of unique environments and tasks, significantly advancing the scope for developing adaptive and generalizable RL agents. This toolkit, inspired by DeepMind's XLand projects, offers an accessible platform for researchers to experiment with and extend RL frameworks, providing essential resources like pre-generated terrains and tasks along with reinforcement learning baselines

---

> ### Author Rebuttal · Authors · 2024-08-17
>
> Thank you for your encouraging feedback on our submission. Your highlighting of the impact of our contributions and their potential to advance the scope for developing adaptive generalizable RL agents is heartening. Pointing out the practical value that this provides to the RL community is particularly gratifying. Below, we reply to the points you raised.
>
> > The paper lacks detailed performance comparisons between U3 and other existing RL frameworks. Including such comparisons would provide a clearer picture of U3's advantages and limitations relative to current tools.
>
> We acknowledge the suggestion to include detailed comparisons to other existing RL frameworks. To include for these, we will expand on the performance comparison to XLand 2.0[1] and XLand-Minigrid[2]. In particular, we note that the training of the Ada model, which required 64 two-core TPUv3 devices running to train, reached approximately 60 steps per second (SPS) per core, whereas the original XLand environment allows for 445 SPS, although with simpler, smaller models. Time allowing before the camera-ready deadline, we will explore the relative SPS of training runs vs environment runs to better understand how these trade off.
>
> > While the paper discusses the general benefits of U3, it does not delve deeply into potential limitations or challenges in specific use cases. For instance, the scalability of U3 with very high-dimensional observation spaces or its performance with extremely complex tasks could be explored further.
>
> Thank you for highlighting this valuable potential contribution. To expand upon this, we explore the impact of environment size on the generation speed. We find that increasing both environment size and production rules by multiple orders of magnitude has a negligible effect, results that we summarize in Table 1, which can be found in the one-page PDF.
>
> > Although the paper provides a good overview of U3's features, it could benefit from a deeper technical discussion on the underlying algorithms and design choices, particularly in the procedural generation processes and their impact on RL training.
>
> We are grateful for your suggestion to expand the discussion on the underlying design choices, in particular with regards to the generation processes and their impact on RL training. In this work, we try our best to remain faithful to the generation processes put in place for the XLand environment space. That is why we base our world generation on the Wave Function Collapse Algorithm[3]. Our production rule generation process also follows a similar pattern to that put forth in XLand 2.0, first generating a set of production rule objects, then generating a chain of rules that ends in reward to ensure task solvability, followed by the addition of dead-end rules. Should this work get accepted, we hope that we will get the opportunity to explore the impact of different environment design choices on RL training along with the community.
>
> > The paper would benefit from a more detailed description of the reward composition mechanisms, which are crucial for understanding how different behaviors are incentivized within the reinforcement learning environments created using the U3 toolkit.
>
> Thank you for pointing out this missing aspect of our text. At line 208, we now include a description of environment details, including the reward mechanism as follows: "In the simulated environment, the agent perceives a first-person visual representation of its surroundings. The agent is capable of executing both continuous and discrete actions. [...] **The agent is positively rewarded with a reward of +1 for each time step during which the predefined goal state is met.** Each trial concludes after a user-specified number of steps, and the episode terminates following a user-defined number of trials."
>
> > Does the paper address whether the consistency of the production rules in the U3 toolkit is checked, and if so, how is this consistency ensured?
> > Does the paper discuss whether the sequence of production rules is verified to ensure that the tasks in the procedurally generated environments are solvable?
>
> Thank you for pointing out these missing details from the paper. The generation process guarantees that the tasks are solvable as follows: we first generate a chain of production rules that enact transformations from the initial state to the reward state. In particular, the conditions that trigger these transitions are forced to be mutually exclusive, ensuring that there exists a valid sequence of actions leading to the reward, even if all production rules are all active at the same time. To make this clear to readers, we will add the following line to the start of the paragraph that begins at line 246: "To ensure a consistent set of production rules that allow for a valid solution, we first sample production rules with mutually exclusive conditions based on the initial objects, such that there is always a valid path to the reward. We subsequently sample “distractor” rules that branch off from the main production rule".
>
>
> > The Limitations section is well written although there are some weak points of this approach missing.
>
> We acknowledge the reviewer's suggestion to include more details in the limitations section. In particular, we will highlight key aspects of performance in the limitations section, including the current steps per second per core of the environment, as well as the expected compute requirements for training an agent similar to Ada.
>
> [1] Team, Adaptive Agent. "Human-timescale adaptation in an open-ended task space." arXiv:2301.07608 (2023).
>
> [2] Nikulin, Alexander et al. "XLand-minigrid: Scalable meta-reinforcement learning environments in JAX." arXiv preprint arXiv:2312.12044 (2023).
>
> [3] M. Gumin. Wave Function Collapse Algorithm, 2016. URL https://github.com/mxgmn/ WaveFunctionCollapse.

---

### Official Review · Reviewer_2JpX · 2024-07-03
**Open-Source and Flexible Procedural Environment Generation in Unity**

**Rating:** 8
**Confidence:** 4
**Correctness:** Yes
**Clarity:** Yes

**Review:**

- Pros
    - The paper was clearly written, making it easy to understand the original XLand approach and how this paper implements an open-source variant.
    - The authors provide an illustrative example of how to use U3 for both the OpenXLand and MemoryMaze domains
    - Many tables are provided to help readers understand what the components of OpenXLand are (Table 1 & 2, Figure 6)
    - The study on how parallelizable U3 (Section 4.2) is very informative, since training on batched environments is common in RL

- Cons
    - There could be a complete figure that illustrates how all of the components connect to form a new environment. It isn't entirely clear to me  how everything fits together.
    - In addition to the analysis on steps per second (SPS), it would be useful to know how much RAM it took to run U3, and if that could limit the complexity of environments that can be generated. This is because procedural generation of assets in Unity could quickly become computationally expensive.

**Strengths:**

See above.

**Additional Feedback:**

### Questions
- Line 112:
> Although production rules do not allow conjunctions
What does conjunction mean in this context? I thought it was possible to use "if" statements in the production rules?

**Documentation:**

Yes, this was described in the supplementary material.

**Limitations:**

Yes, the authors discussed limitations in Section 6.

**Opportunities For Improvement:**

- Lines 71-73:This could be clarified since Minecraft is inherently procedurally generated, even without Minedojo.
- It could be useful to mention that Unity is available free for personal use, so that interested members of the community can try out U3: https://unity.com/products/unity-personal.
- I think it may help readers understand prior work better if there were separate paragraph headers in the related work section. For example, one paragraph could be “meta-RL”, then “procedurally generating environments for RL”, and “using game engines for generating environments”.
- Lines 112-114: It would be nice to have a large figure in the main text that shows the world, goal states, and production rules. The authors could potentially remove Figure 1 to make room for this.

**Relation To Prior Work:**

Yes

**Summary And Contributions:**

- This paper presents United Unity Universe (U3), which is a platform built in Unity for procedurally generating reinforcement learning environments. By incorporating modular and compositional components such as production rules and metatiles, users are able to implement a wide variety of environments.

- The authors use U3 to reproduce XLand, then release datasets of generated worlds and production rules, which can then be combined to create new MDPs (environments).

---

> ### Author Rebuttal · Authors · 2024-08-17
>
> Thank you for your thoughtful review of our paper. We are glad you found our presentation clear and appreciated the illustrative examples and tables. Your constructive suggestions for improvement are invaluable, and we will address them in our revisions. Below, we respond to each point
>
>
> > In addition to the analysis on steps per second (SPS), it would be useful to know how much RAM it took to run U3, and if that could limit the complexity of environments that can be generated. This is because procedural generation of assets in Unity could quickly become computationally expensive.
>
>
> Thank you for suggesting this valuable clarification that will help potential users understand the environment better and maximize the value they can get from using U3. In our training, we found that the memory buffers of our RL algorithms were using more memory than the environments themselves.  However, to be thorough, we will explore this issue in more detail. In the camera-ready version, we will include RAM requirements across different environment complexities – similar to Table 1 in the one-page PDF.
>
>
> > There could be a complete figure that illustrates how all of the components connect to form a new environment. It isn't entirely clear to me how everything fits together.
> > Lines 112-114: It would be nice to have a large figure in the main text that shows the world, goal states, and production rules. The authors could potentially remove Figure 1 to make room for this.
>
>
> We acknowledge the reviewer's insightful proposition of a figure that would help better convey an overview of U3. In the one-page PDF, we have included such a proposed figure that illustrates the generation of a world and of a task, including its goal state, production rules, and a graph of possible transitions, to convey how all the elements coalesce into a new environment. This figure would be placed directly below Figure 1, as we believe these figures complement each other by the first painting a high-level overview of U3, and the second grounding U3's capabilities with the concrete example of OpenXLand. We would greatly appreciate your feedback to strengthen this figure's overview of the environment.
>
>
> > Line 112: Although production rules do not allow conjunctions What does conjunction mean in this context? I thought it was possible to use "if" statements in the production rules?
>
>
> We apologize for the lack of clarity in our original explanation. In the original Ada paper, the authors specify in appendix A.1., bullet point 3, that their production rules use only a single condition. They do this by expressly excluding production rule conditions with logical combinations, such as conjunctions, e.g. ``red sphere near agent AND blue sphere near red sphere``, or disjunctions, e.g. ``red sphere near agent OR blue sphere near red sphere``. However, we note that U3 does not possess this limitation, and can indeed implement production rules with logical operators connecting them, such as is the case in our OpenXLand implementation. To rectify this, we amend the text as follows starting at line 112: ~~Although~~ **We note that** production rules **in XLand and XLand 2.0** do not allow conjunctions, **e.g. ``red sphere near agent AND blue sphere near red sphere``, or disjunctions, e.g. ``red sphere near agent OR blue sphere near red sphere``.**  “Predicate” refers to both the goal state and the production rule conditions. Together, the agents, the world, goal states and production rules for XLand 2.0 [1] make up a task.
>
>
> > Lines 71-73:This could be clarified since Minecraft is inherently procedurally generated, even without Minedojo.
> > It could be useful to mention that Unity is available free for personal use, so that interested members of the community can try out U3: https://unity.com/products/unity-personal.
> > I think it may help readers understand prior work better if there were separate paragraph headers in the related work section. For example, one paragraph could be “meta-RL”, then “procedurally generating environments for RL”, and “using game engines for generating environments”.
>
>
> Thank you for mentioning these clarifications. For the camera-ready version, we plan to include the following corrections:
> - Lines 71-73: Malmo creates an API for Minecraft to enable RL training **in its procedurally generated 3D worlds**, while MineRL expands the API to leverage datasets of pre-existing content ~~and Minedojo expands this effort, notably with procedural generation~~**, an effort that Minedojo considerably expands.**
> - Lines 23: Inspired by the use of the Unity Game Engine for reinforcement learning development, we propose the *United Unity Universe* (U3), an ~~Unity-based~~ open-source environment development ecosystem based on the Unity game engine, which is freely available for academic use.
> - We have added separate headers to the paragraphs of the related work: **Meta-Reinforcement Learning, Procedural Generation in Reinforcement Learning Environments, The XLand Environment Spaces**
>
> [1] Team, Adaptive Agent. "Human-timescale adaptation in an open-ended task space." arXiv:2301.07608 (2023).

---

### Official Review · Reviewer_gZM4 · 2024-07-19
**An exciting new framework for RL environments**

**Rating:** 7
**Confidence:** 5

**Review:**

Overall OpenXland and U3 seem like a valuable new benchmark for the RL community. The U3 framework can be used to develop many new complex environments, as demonstrated by their reimplementation of Xland 2.0. The RL community has been lacking an environment similar to XLand for along time, and this work may help to finally produce an open-source recreation of Ada. By providing large datasets of pre-generated worlds and rules, this paper goes a long way toward promoting the use of their framework. Its clear that this work will be a valuable resource to the community

Although the contributions have clear value, this paper lacks some experiments and discussion that are typical for RL benchmark papers. It would be helpful if this paper included an evaluation protocol to ensure that future work is comparable, especially in a meta-learning environment where the protocol is not clear. Section 5.2 includes only a short paragraph describing their evaluations used for their experiments, which references "trials" without explaining what that means in their environment. The authors might also consider creating separate out of distribution evaluation sets of production rules and maps. Similarly, this paper does not provide a competent baseline for OpenXLand. They train a SAC agent for 75 million steps to demonstrate that the environment is learnable, but without analyzing the agent's behavior, its impossible to tell whether it is learning intelligent behavior or reward hacking. The paper also lacks many important environment details, such as the reward function structure, action space, observation space, termination conditions, etc. Some of these issues would require time and computational resources to solve, but it should be possible to add in missing environment details or suggest an evaluation protocol.

**Strengths:**

I stated some of the strengths of this work in the overall review, but to summarize:
* Flexible, extensible framework for creating complex environments that takes advantage of the Unity engine to produce visually interesting, challenging tasks.
* An open source reimplementation of XLand 2.0, which could enable a full reproduction of Ada, one of the most promising meta-RL algorithms.
* Entirely open-sourced codebase and datasets available on Huggingface.

**Additional Feedback:**

It can be difficult to adopt a new environment, especially one as complex as this, without a strong baseline to start with. Not only can developing the first baseline take significant effort and computational resources, but it often reveals major bugs in the environment. Most of the research on NetHack has been directly built on top of the original baseline introduced by the paper, and then models developed in competitions. Similarly the NeuralMMO baselines were mostly developed through competitions at ML venues, starting with a baseline provided by the organizers. Maybe even before replicating Ada, which is a very complex algorithm, it may be worth trying to train a competent SAC agent that can demonstrate some basic skills and understanding of the environment.

**Clarity:**

In addition to missing some details about the environment, it is not clear what the distinction is between Unity's ML Agents, United Unity Universe, and OpenXLand. In particular it's not obvious how ML agents and U3 are distinct, which makes it difficult to judge its novelty. My interpretation is that Unity ML-Agents provides an interface between Unity and python for creating RL agents, and U3 extends this adding additional communication layers, a procedural generation system including metatiles, and production rules. OpenXLand 2 is effectively a specific collection of metatiles and production rules that mimic the features in XLand 2.0. On line 39 the paper mentions "A two-way pipeline connecting U3 and Python for efficient training, agent evaluation 40 and runtime adaptive auto-curricula" and later expands on this slightly in Section 4.4. However, Unity's ML-Agents claims to include flexible communication between a python trainer and Unity RL environment, so it's unclear what specifically U3 adds. Regardless of this, it is obvious that OpenXLand is unique from any previously developed open-source environment.

One question, are the steps per second listed in Figure 2 measured during training or only running the environment independently?

There are a couple formatting issues in the paper as well:
* Appendix C lines 514 and 515 seem to have extra numbers.
* Figure 5 seems to have many more legend items than necessary. It seems that there is a small sliver representing certain datasets. I'm not sure if this is an error or if maybe there's a better way of presenting the data?
* Appendix A.1.2 has some weird formatting

**Correctness:**

It would be difficult to assess the correctness of the code, and the experiments do not provide convincing evidence that the environment works entirely as intended, but any issues will hopefully be found with continued development and future research using OpenXLand. The paper's claims about the datasets are supported by the figures.

**Documentation:**

There is some documentation of U3 and OpenXLand in their Github repo. The datasets are documented in the paper and supplementary materials.

**Ethics:**

I do not suspect any ethical concerns.

**Limitations:**

As the authors point out, their recreation of XLand is missing a few features. The work also does not present any evidence that it is possible to train capable agents in the environment, only that there is some learnable signal. The environment may therefore have some bugs, especially with this many complex systems. The environment also runs much slower than other popular benchmarks making it difficult to use for academic research.

**Opportunities For Improvement:**

This work would surely benefit from providing a stronger baseline, either through longer training or hyperparameter tuning. At the very least, running a few more seeds of the same agent would help future researchers to understand what variance to expect between training runs. I would also like to see more details about the environment, a proposed evaluation protocol, and clarity about the difference between U3 and ML-Agents. All of these changes would go a long way towards promoting adoption of the environment.

**Relation To Prior Work:**

Overall the paper does a good job referencing prior relevant work. It does seem to be missing any comparison to Neural MMO 2.0 [1], which is arguably the most relevant open-source environment. It includes a compositional, predicate task system that was also modeled after XLand 2.0's task system.

[1] Suárez, Joseph, et al. "Neural MMO 2.0: a massively multi-task addition to massively multi-agent learning." Advances in Neural Information Processing Systems 36 (2024).

**Summary And Contributions:**

This paper introduces U3, a framework for developing RL environments in Unity, and OpenXLand, an open source reimplementation of XLand2 using U3. They develop a procedural generation system using metatiles and wave function collapse, and allow for production rules which can be used to combine objects in the environment. They use these to reimplement XLand2, a complex multi-task environment developed by DeepMind that was never publicly released. To support future work on OpenXLand, the authors also provide multiple datasets of maps and production rules, separated according to difficulty. The paper demonstrates that the datasets do in fact have different distributions of complexity, that the environment scales with more CPUs, and that the environment is learnable with an SAC agent.

---

> ### Author Rebuttal · Authors · 2024-08-17
>
> We are very grateful for your thorough and insightful feedback on our submission. We are encouraged by your recognition of the strengths of our contributions, such as the flexibility that U3 affords. It is particularly uplifting that you recognize the value of our contributions to the RL community and their potential to help recreate Ada. Please note that our terse responses to your insightful reviews are due to the 6000 character limit:
>
> > It would be helpful if this paper included an evaluation protocol to ensure that future work is comparable, especially in a meta-learning environment where the protocol is not clear.
>
> We acknowledge the need for an evaluation protocol. As such, we propose one based on [1, 2] in the global rebuttal.
>
> > Section 5.2 [...] references "trials" without explaining what that means in their environment.
>
> Thank you for catching this oversight. We will introduce the definition in the background, where most of the nomenclature is introduced.
>
> > The authors might also consider creating separate out of distribution evaluation sets of production rules and maps.
>
> We acknowledge this valuable suggestion to include OOD datasets. We propose to implement the missing "seeing" production rule action, as well as new objects, to generate datasets with objects and rules unseen during training.
>
> > [...] without analyzing the agent's behavior, its impossible to tell whether it is learning intelligent behavior or reward hacking.
>
> We acknowledge the reviewer's insightful comment about the ambiguity of the current environment's state. We propose to implement tooling to analyze agent behavior, such as a heatmap of paths taken in the environment.
>
> > The paper also lacks many important environment details, such as the reward function structure, action space, observation space, termination conditions, etc.
>
> Thank you for these valuable suggestions. Our proposed evaluation protocol is described in the global rebuttal. We propose to include an exhaustive cataloguing of actions and observations, as in appendix C.2 of [1]. We will also include reward details (1 for each time step in which the goal state is met), and termination conditions for trials (number of steps) and episodes (number of trials).
>
> > [...] running a few more seeds of the same agent would help future researchers to understand what variance to expect between training runs.
>
> Thank you for this insightful comment. We are currently running additional seeds, which should complete shortly as we are compute-constrained, and we will report back.
>
> > The environment may therefore have some bugs, especially with this many complex systems.
>
> Thank you for this valuable feedback. We hope that the tooling we mentioned will provide some clarity on the learnability of the environment, and that, as the reviewer suggested, additional development will bring to light any outstanding issues.
>
> > The environment also runs much slower than other popular benchmarks making it difficult to use for academic research.
>
> Thank you for this comprehensive discussion surrounding performance. In the camera-ready version, we will add a discussion to our limitations to present what practitioners can expect in terms of steps per second, clarify Ada's compute requirements and make explicit the main drivers of increased computational burden.
>
> > In addition to missing some details about the environment, it is not clear what the distinction is between Unity's ML Agents, United Unity Universe, and OpenXLand.
>
> Thank you for this constructive comment. To highlight the differences between U3 and ML-Agents [4], we amend the discussion in lines 204-206 to clarify that U3 utilizes ML-Agents to provide a two-way communication channel between Unity and Python. Specifically, U3 provides a protocol for high-level environment manipulation by defining the parameters of the environment generation process." We also hope that the figure in the one-page PDF, which we will place below the current Figure 1, will clarify the relationship between U3 and OpenXLand.
>
> > One question, are the steps per second listed in Figure 2 measured during training or only running the environment independently?
>
> These are run independently, but we achieve a comparable ~500 steps per second across 16 cores when training the SAC agent. As a reference, ML-Agents reaches ~108 SPS (9.23 milliseconds per step) on 84x84x3 observations. We therefore believe there is room for improvement.
>
> > There are a couple formatting issues in the paper as well:
>
> Thank you for attentively reviewing our submission. We will fix the formatting in the appendices, and rework the caption of Figure 5 to be more clear.
>
> >It does seem to be missing any comparison to Neural MMO 2.0 [1], which is arguably the most relevant open-source environment.
>
> Thank you for suggesting this important and highly relevant work. We will compare and contrast neural-MMO 2.0 in our related work. In addition, we will also discuss the limitations to the accessibility of U3 for academic research as compared to other environments, such as neural-MMO 2.0 and XLand-minigrid.
>
> > Maybe even before replicating Ada, which is a very complex algorithm, it may be worth trying to train a competent SAC agent that can demonstrate some basic skills and understanding of the environment.
>
> Thank you for the insightful comment about the importance of a strong baseline for environment adoption. Unfortunately, we are compute-restricted as it stands, so we are unlikely to produce a strong baseline before the end of discussions. However, we will still attempt to train a moderately competent SAC agent as a baseline before the camera-ready deadline.
>
> > Some of these issues would require time and computational resources to solve, but it should be possible to add in missing environment details or suggest an evaluation protocol.
>
> Thank you for the thoroughness of your review, and the constructiveness of your feedback. We hope this work provides value to the RL community.

---

> > ### Author Response · Authors · 2024-08-17
> > **Citations**
> >
> > Apologies for the oversight.
> >
> > Here are the accompanying citations:
> >
> > [1] Team, Adaptive Agent. "Human-timescale adaptation in an open-ended task space." arXiv:2301.07608 (2023).
> >
> > [2] Nikulin, Alexander et al. "XLand-minigrid: Scalable meta-reinforcement learning environments in JAX." arXiv preprint arXiv:2312.12044 (2023).
> >
> > [3] Suárez, Joseph, David Bloomin, Kyoung Whan Choe, Hao Xiang Li, Ryan Sullivan, Nishaanth Kanna, Daniel Scott et al. "Neural MMO 2.0: a massively multi-task addition to massively multi-agent learning." Advances in Neural Information Processing Systems 36 (2024).
> >
> > [4] Juliani, Arthur et al. "Unity: A general platform for intelligent agents." arXiv preprint arXiv:1809.02627 (2018).

---

> > ### Comment · Reviewer_gZM4 · 2024-08-22
> >
> > Thank you for the detailed response, I think the proposed changes will greatly improve the impact of your work. I'm looking forward to hearing more about this project in the future.

---

> > > ### Author Response · Authors · 2024-08-28
> > >
> > > Thank you for suggesting the changes, we believe this work will greatly benefit from them.

---

### Official Review · Reviewer_XKFT · 2024-07-25

**Rating:** 6
**Confidence:** 3
**Correctness:** The claims made by the paper seem cor…
**Clarity:** The paper is well-written.

**Review:**

The framework proposed by the authors serves to tackle an important gap. The paper is well-written with a few shortcomings, for example, the discussion of the performance of the implemented method is not comprehensive.

**Strengths:**

* The paper is clearly written with a well-explained motivation.

* The paper introduces a toolkit U3 and a corresponding dataset to help further developments.

**Additional Feedback:**

Using Unity could increase the complexity of training of RL and the inefficiency of training. If the authors could discuss this point in detail it would improve our understanding of both the strengths and limitations of their proposal.

**Documentation:**

Yes.

**Ethics:**

No.

**Limitations:**

The author discusses the difficulty of setting up the system and some future updating planning in the limitation section.

**Opportunities For Improvement:**

The discussion of the performance of the implemented method should be made more comprehensive.

**Relation To Prior Work:**

The authors included a comprehensive discussion of prior work and describe some of the shortcomings of existing methods. They also describe the relationship between their ideas and prior works.

**Summary And Contributions:**

The paper presents the United Unity Universe (U3), an open-source toolkit built on top of the Unity game engine. U3 includes a robust implementation of OpenXLand, a meta-RL framework, and offers a user-friendly interface for modifying procedural terrains and task rules. It also provides a curated selection of terrains, rule sets, and implementations of RL baselines to facilitate experimentation and development of adaptive agents.

---

> ### Author Rebuttal · Authors · 2024-08-17
>
> We thank you for highlighting the impact of our paper. We hope that our framework can help to fill some of the gap between commercial organizations and the open-source community. Your insightful comments will bring us closer to this goal. We address the points raised below:
>
> > The paper is well-written with a few shortcomings, for example, the discussion of the performance of the implemented method is not comprehensive.
>
> > Using Unity could increase the complexity of training of RL and the inefficiency of training. If the authors could discuss this point in detail it would improve our understanding of both the strengths and limitations of their proposal.
>
> Thank you for pointing out this missing discussion on the inefficiency of training due to using Unity. We believe that discussing such issues will strengthen our text and help scope out the accessibility of the environment so that prospective users can understand what the increased overhead over other environments enables them to gain access to. To rectify this, we will add two things to the camera-ready version: a **discussion on the performance**, as well as expected resource requirements in the limitations for those who would like to use this environment. We will include a comprehensive discussion of the impact of using Unity on performance, in particular the impact on steps per second (SPS) when compared to other environments such as XLand 2.0 [1], XLand-minigrid [2] and NeuralMMO 2.0 [3]. Secondly, we will include a table (Table 1 in the one-page PDF) which explores **the effect of performance on environment complexity**. We will also discuss the compute requirements for the Ada architecture to establish realistic expectations for practitioners looking to use this environment for RL training.
>
> [1] Team, Adaptive Agent. "Human-timescale adaptation in an open-ended task space." arXiv:2301.07608 (2023).
>
> [2] Nikulin, Alexander et al. "XLand-minigrid: Scalable meta-reinforcement learning environments in JAX." arXiv preprint arXiv:2312.12044 (2023).
>
> [3] Suárez, Joseph, David Bloomin, Kyoung Whan Choe, Hao Xiang Li, Ryan Sullivan, Nishaanth Kanna, Daniel Scott et al. "Neural MMO 2.0: a massively multi-task addition to massively multi-agent learning." Advances in Neural Information Processing Systems 36 (2024).

---

### Author Rebuttal · Authors · 2024-08-17

We gratefully acknowledge all of the reviewers for their insightful comments. We appreciate that reviewers recognize the quality of our presentation (reviewers XKFT, 2JpX, Trj1), the value of our proposed U3 framework's versatility (reviewers XKFT, gZM4, Trj1) and scalability (reviewers Trj1), and the additional benefit of pre-generated implementations and datasets (reviewers XKFT, gZM4, 2JpX, Trj1), including OpenXLand and MemoryMaze. Below, we review major changes made following the reviewers' constructive feedback.


**Performance**


As noted by all reviewers, the environment's **performance**, including factors such as scalability, parallelizability, and steps per second (SPS), is a key aspect impacting the complexity and inefficiency of training (reviewer XKFT), which has a downstream impact on the environment's accessibility for academic research (reviewer gZM4).


Indeed, the original single-agent Ada agent took 5 weeks of 64 two-core TPUv3 specialized devices to train their single-agent model for 25 billion steps, reaching a step rate of approximately 65 SPS across all 128 cores. The inefficiency of this environment makes it  very demanding in terms of compute compared to what optimized RL environments, such as Neural-MMO, can reach on a single core.


In our experiments, our implementation reaches 31.25 SPS per core during training (500 SPS across 16 cores). Although our environment is not yet as fast as DeepMind's implementation, we believe there is significant room to optimize U3's performance in terms of SPS per core down the line. In fact, the original ML-Agents [3] package reaches approximately 100 SPS for a 3D environment on unspecified hardware.  This could lead to a high overall SPS when scaled across more cores.


We are grateful that the reviewers pointed out that the performance of the environment, as well as the expected requirements to train capable agents, are important factors for  efficiency and accessibility, and as such should be discussed in more detail in the text. Therefore, we implement the following modifications:


- We expand the technical discussion of performance with **new measures of scalability in the size of the environment and the complexity of the tasks**. We will also add measures of **resource usage**, including RAM.
- We also highlight key aspects of performance in the limitations section, including the **SPS and the expected compute requirements** for training Ada.
- We add the **optimization of the SPS of the environment** to future work.
- In the broader impact statement, we discuss the issue of **accessibility** due to compute requirements, including pointers to environments that are more accessible to academic researchers, such as Neural-MMO and XLand-minigrid.


**Evaluation protocol**


We acknowledge the need for an evaluation protocol pointed out by reviewer gZM4 to ensure comparability of future work. As such, we propose the following:


- For each of the six datasets, we propose a holdout set of 4096 tasks, as in XLand-Minigrid [1], with 8 trials each, as this is the number of trials for which a trained Ada obtains a last-trial performance close to that of an agent specifically fine-tuned to a task [2].
- We propose to compare performance at the 20th percentile across tasks, as this avoids simpler tasks dominating the average performance as pointed out in [1] and [2].
- Furthermore, we propose for now to restrain method comparison to within datasets, as this avoids costly normalization by compute-heavy fine-tuned baselines.


To describe this evaluation protocol, we will include a new subsection in section 5.


**Other corrections**


Below, we list a summary of other corrections and additions:


- We **add an additional figure to the second page of the paper** which gives a comprehensive overview of how different components such as metatiles, goal states and production rules come together to form a new environment.
- We include **additional details on the differences between ML-Agents and U3**, focusing on ML-agents as the low-level communication protocol, and U3 as the high-level world building framework.
- We will expand our discussion of the procedural generation processes by **adding a table showing the amount of time taken to build each dataset**. As a rough estimate, the 1 million environment datasets took ~10 minutes (0.6ms per env) to generate for the Easy (smallest) datasets, ~2 hours (7.2ms per env) for the Medium datasets and ~ 2 days (172.8ms per env) for the Hard (largest) datasets.
- We clarify why the production rule generation process generates **consistent, solvable sets of production rules**.
- We **add key environment details**, such as the reward structure, the action space, the observation space, and termination conditions.
- We introduce the concept of ‘**trials**’.
- We clarify how the original Ada team uses the term ‘**conjunction**’ to refer to logical operators between production rule conditions.
- We mention that **Unity is available free for academic use**.
- We **add Neural-MMO to related work**, as well as highlighting its **accessibility for academic research** in the broader impact statement.
- We fix **formatting issues**, including lines 514-515 in appendix C, Figure 5's legend, Appendix A.1.2, Minecraft's inherent procedural generation on lines 71-73, separate headers in the related work section.


[1] Nikulin, Alexander et al. "XLand-minigrid: Scalable meta-reinforcement learning environments in JAX." arXiv preprint arXiv:2312.12044 (2023).


[2] Team, Adaptive Agent. "Human-timescale adaptation in an open-ended task space." arXiv:2301.07608 (2023).

[3] Juliani, Arthur et al. "Unity: A general platform for intelligent agents." arXiv preprint arXiv:1809.02627 (2018).

---

### Decision · Program_Chairs · 2024-09-26

**Decision:**

Accept (Poster)

**Comment:**

In the manuscript, the authors propose the U3 (United Unity Universe) framework that can generate reinforcement learning environments. The authors demonstrated the use of U3 by implementing OpenXLand and MemoryMaze.

Strengths:
1. The reviewers generally found U3 to be a useful open-source framework due to its flexibility and extensibility.
2. The implementation of OpenXLand and MemoryMaze provided concrete demonstrations of the use of U3.
3. The manuscript was well written in general.

Weaknesses:
1. Most reviewers asked questions about the environment's performance, which the original version of the manuscript does not provide sufficient information about.
2. There were questions about the details of the environments.
3. There were also questions about evaluation protocols in the benchmark.

The authors provided rebuttals and revised the manuscript based on the reviewers' comments. One of the four reviewers confirmed that the changes proposed by the authors would be useful. All four reviewers gave a score above the acceptance threshold and one of the reviewers who gave the lowest score (6) had the lowest confidence level. Considering all these, the Area Chair concludes that there was a general enthusiasm about this work among the reviewers.